# Advanced Evaluation of the Freeze–Thaw Damage of Concrete Based on the Fracture Tests

**DOI:** 10.3390/ma14216378

**Published:** 2021-10-25

**Authors:** Barbara Kucharczyková, Hana Šimonová, Dalibor Kocáb, Libor Topolář

**Affiliations:** Faculty of Civil Engineering, Brno University of Technology, Veveří 331/95, 602 00 Brno, Czech Republic; simonova.h@vutbr.cz (H.Š.); dalibor.kocab@vutbr.cz (D.K.); libor.topolar@vutbr.cz (L.T.)

**Keywords:** freeze–thaw, fracture, toughness, energy, double-K, crack extension, crack opening, acoustic emission, RMS

## Abstract

This paper presents the results of an experimental program aimed at the assessment of the freeze–thaw (F–T) resistance of concrete based on the evaluation of fracture tests accompanied by acoustic emission measurements. Two concretes of similar mechanical characteristics were manufactured for the experiment. The main difference between the C1 and C2 concrete was in the total number of air voids and in the A300 parameter, where both parameters were higher for C1 by about 35% and 52%, respectively. The evaluation of the fracture characteristics was performed on the basis of experimentally recorded load–deflection and load–crack mouth opening displacement diagrams using two different approaches: linear fracture mechanics completed with the effective crack model and the double-K model. The results show that both approaches gave similar results, especially if the nonlinear behavior before the peak load was considered. According to the results, it can be stated that continuous AE measurement is beneficial for the assessment of the extent of concrete deterioration, and it suitably supplements the fracture test evaluation. A comparison of the results of fracture tests with the resonance method and splitting tensile strength test shows that all testing methods led to the same conclusion, i.e., the C1 concrete was more F–T-resistant than C2. However, the fracture test evaluation provided more detailed information about the internal structure deterioration due to the F–T exposure.

## 1. Introduction

Concrete belongs to the most common building materials used in various civil engineering applications. The world produces about 4.4 billion tons of concrete annually, of which a substantial portion is consumed for the construction of transportation networks with strict requirements for their durability. This means that, in addition to basic strength and deformation parameters, the characteristics related to water, air permeability, and cracking tendency are strictly monitored under different weather conditions.

The alternation of positive and negative temperatures (freeze–thaw cycles) is considered one of the most destructive processes that substantially influence the durability of concrete structures [1]. It can be stated that the freeze–thaw (F–T) resistance of concrete expresses the resistance of the concrete to the interaction of all physical, mechanical, and fracture processes that act together at one moment. The rate of the deterioration of the concrete structure depends on the number of F–T cycles, as well as on the absolute values of alternating temperatures.

There are various testing approaches and procedures (direct and indirect) for the assessment of the F–T resistance under laboratory conditions, which are adjusted to the actual weather conditions of a particular world region or country, and they mainly differ in the length of the F–T cycle, temperature range, monitoring intervals, monitored characteristics, and the limit number of F–T cycles or the limit value prescribed for a decrease in the monitored characteristic [2]. The relative changes in compressive and tensile strength or dynamic modulus of elasticity, monitored as a function of changes in mass and length, are the most common parameters used for examination of the macroscopic performance of concrete materials exposed to F–T cycles [1]. Although researchers have indicated that the fracture behavior is more sensitive to F–T damage, the fracture tests are performed rather rarely and the fracture behavior is, in this case, still frequently neglected [3].

There are reasonable arguments for why the fracture characteristics should be suitable for the assessment of the rate of F–T deterioration. They arise from the essence of the fracture theory which deals with the resistance of the material to crack initiation and propagation [4]. Fracture toughness and fracture energy are basic and the most commonly used parameters to express the fracture behavior of the materials. The F–T resistance is mostly examined through the variations in the value of fracture energy or fracture toughness [5,6,7].

The F–T resistance of concrete depends on the quality of components and the overall composition of the fresh concrete, as well as on the quality of compaction and curing after its placing. In the hardened state, the F–T resistance depends on the quality of the cement matrix, aggregate, and interfacial transition zone (ITZ) between the aggregate and cement matrix. The air-entraining admixture is commonly used for enhancing the F–T resistance of concrete [8]. In this case, the air voids are intentionally spread in fresh concrete to obtain a uniform air-void system, containing a predefined minimum of voids smaller than 300 microns in the hardened concrete. It was observed that the presence of closed air voids influences the evolution of the fracture damage, which differs compared to ordinary non-air-entrained concrete, especially in the coalescence of voids and cracks [8,9].

The main objective of this paper was to present the specifics, advantages, and disadvantages of the fracture tests performed in the context of F–T damage to the professional and general public. A wide range of fracture parameters were evaluated in the paper to discuss the sensitivity of a particular parameter to the extent of concrete deterioration due to F–T cycling. In addition, the results of acoustic emission (AE) measurements during the fracture tests, which is a tool for nondestructive monitoring of active changes in progress during the loading of concrete specimens, are presented. These results were compared with the results of the commonly used testing approaches, i.e., changes in fundamental resonant frequency, mass, compressive strength, and tensile strength.

## 2. Materials and Methods

Two types of ordinary concrete, C1 and C2, with natural aggregate were designed for the experiment. The composition of both concretes per 1 m^3^ is given in Table 1.

The ready-mix concretes were prepared in a concrete plant and supplied to the laboratory for molding the test specimens. A combination of two plasticizing admixtures Sika ViscoCrete-4035 (superplasticizer with a fast effect) and Sika ViscoFlow-25 (plasticizer with delayed effect and stabilizing character) [10] was used to retain the workability and stability of ready-mix concrete for a longer time, as needed for the transportation and molding of quite a large number of the test specimens. Prismatic specimens with nominal dimensions of 100 mm × 100 mm × 400 mm were prepared and, after demolding, were cured in the water bath until the time of testing. In both cases, the age of the test specimens at the start of the F–T tests was at least 90 days when the strength characteristics were already stabilized. In total, 12 test specimens were prepared from each concrete for the fracture tests. In both cases, three test specimens served as reference (non-frost-attacked) specimens, while the remaining specimens were subjected to the F–T cycles.

### 2.1. Freeze–Thaw Test

The F–T tests were performed according to the standard ČSN 73 1322 [11] valid for the testing of the F–T resistance of concrete in the Czech Republic. This procedure specifies the F–T cycles within the temperature range from −18 °C to +20 °C. Each F–T cycle consists of 4 h of air-freezing and 2 h of thawing in the water bath, see Figure 1(c).

An automatic freeze–thaw cabinet KD 20 (manufactured by EKOFROST s.r.o., Olomouc, Czech Republic) was used for the experiment, see Figure 1. In this particular case, the interval for nondestructive monitoring and for the measurement of dimensions and mass of the test specimens was set to 25 F–T cycles. The fracture tests were performed after 0 (non-frost attacked), 50, 100, and 200 cycles. Each test set contained three test specimens. The total duration of the 200 cycles was 56 days. The reference non-frost-attacked specimens of both C1 and C2 concretes were stored in the water bath until the 50 F–T cycles were finished on the other set of specimens. Then, the reference non-frost-attacked specimens were tested at the same time as the set of specimens subjected to 50 F–T cycles.

### 2.2. Test Method for Fundamental Longitudinal Frequency

A nondestructive test based on the resonance method (see Figure 2) was employed to monitor the development of the dynamic modulus of elasticity *E*_rL_ and dynamic Poisson’s ratio *µ*_r_ of concretes during the F–T test. All specimens were measured before the start of the F–T test. The specimens subjected to the F–T action were measured at regular intervals (after each 25 F–T cycles) throughout the F–T test. The natural frequency of longitudinal and torsional vibrations was measured using a Handyscope HS4 oscilloscope equipped with an acoustic sensor. The readers are referred to [12] for more details about the principle of measurement. The absolute values of *E*_rL_ and *µ*_r_ were calculated in compliance with ASTM C215-19 [13] as follows:$$E_{\mathrm{rL}}=4\frac{L}{WB}mf_{L}^{2},$$
where *E*_rL_ is the dynamic modulus of elasticity, *L* is the length of specimens, *W* and *B* are cross-section dimensions, *m* is the mass of specimens, and *f*_L_ is the fundamental longitudinal frequency.
$$\mu_{r}=\frac{E_{\mathrm{rL}}}{2\cdot G_{r}}-1,$$
where *µ*_r_ is the dimensionless dynamic Poisson’s ratio, and *G*_r_ is the dynamic modulus of rigidity, calculated as
$$G_{r}=4\frac{LR}{WB}mf_{t}^{2},$$
where *R* is the shape factor (1.183 for a square cross-section prism), and *f*_t_ is the fundamental torsional frequency.

### 2.3. Acoustic Emission Method

The AE method is a tool for the nondestructive monitoring of active changes in a material produced during the loading of concrete specimens. The principle of the method consists of the continuous monitoring of the acoustic response caused by crack initiation and propagation during the loading of the specimen. To analyze the extent and progress of the specimen deterioration, it is very important to define an appropriate method for AE signal identification even before the start of the measurement. The most widely used approach is based on the setup of a signal threshold to distinguish failure. This specific approach presumes that each signal exceeding this threshold indicates a certain type of material disruption. The extent of material deterioration can be determined as a function of the number of AE events (counts) or the time of signal duration [14]. However, in case individual AE signals occur successively very close to each other, their separation can be problematic, which leads to errors in their evaluation (e.g. false counts) as reported in [15]. It was proven that, in such cases, the root-mean-square (RMS) value of the AE signal envelope is more effective for evaluation [15]. The RMS is an AE parameter that is proportional to the square root of the quantum of energy transmitted by the AE wave. The RMS value increases with the increasing deterioration of the material during loading [16,17].

In the experiment presented herein, the monitoring of the AE activity was done using a double-channel unit DAKEL ZEDO with the following input parameters: the threshold value for counts was 0.561 mV, the threshold value for individual AE hits was 56 µV, the sampling frequency of AE hits was set to 10 MHz, and the cutoff frequency of the low-pass filter was set to 800 kHz. The total gain was 59 dB (pre-amplifier 34 dB and amplifier 25 dB). The AE sensors were attached to the specimens with beeswax in a thin layer. The extent of specimen deterioration was expressed by the cumulative value of RMS calculated for specific load intervals.

### 2.4. Fracture Test

Before testing, all test specimens were provided with an artificial notch with a depth of approximately one-third of the specimen height using a diamond blade saw. The test specimens were subjected to three-point bending tests (span length was 300 mm) with a constant displacement increment of 0.02 mm/min. This allowed obtaining the whole record of the load–deflection (*F‒d*) and load–crack mouth opening displacement (*F‒CMOD*) diagrams. In all cases, the test was terminated at a deflection of the specimens of at least 0.6 mm (the value of loading force was already close to zero). The surfaces, especially those near the artificial notch, of all test specimens were inspected using a digital microscope with a magnification of 250× to verify the presence of microcracks just before the start of the test.

A multifunctional testing machine LaborTech with a loading range of 250 kN (equipped with an output channel for the loading force), an inductive sensor with a measurement range of 2 mm, a strain gauge, and a Quantum data-taker were used for testing. This apparatus allowed a precise setup of the test and a continual record of all measured quantities. The inductive sensor was used for the measurement of specimen deflection. It was mounted on the front of a special frame bedded on the upper surface of the specimens during the test (see Figure 3). The frame was constructed to measure the deflection of twofold values in the middle of the span length. The strain gauge was used for the measurement of the crack mouth opening displacement (CMOD). It was mounted between two blades glued on the bottom surface of the specimens symmetrically to the artificial notch. The arrangement of the test is shown in Figure 3.

All fractured specimens were further used for the determination of the compressive and splitting tensile strength on specimen fragments.

### 2.5. Evaluation of the Fracture Tests

All the recorded *F‒d* and *F‒CMOD* diagrams were processed using GTDiPS software before their evaluation [18] (refer to [19] for more details).

#### 2.5.1. Fracture Toughness

The fracture toughness value was determined using the linear elastic fracture mechanics approach for brittle fracture. This parameter is related to the stress field near the tip of the crack. The fracture toughness value *K*_Ic_ is calculated as follows [4]:(1)$$K_{\mathrm{Ic}}=\frac{6M_{\max}}{BW^{2}}Y\left(\alpha \right)\sqrt{a},$$
where *M*_max_ is the bending moment due to the maximum load *F*_max_ and self-weight, *B* is the specimen width, *W* is the specimen depth, *Y*(*α*) is a function of geometry [4], and *a* is the initial notch depth.

In this case, a geometry function for three-point bending configuration proposed by Brown and Srawley (1966) was used [4].
(2)$$Y\left(\alpha \right)\;=1.93-3.07\alpha +14.53\alpha^{2}-25.11\alpha^{3}+25.80\alpha^{4},$$
where *α* = *a*/*W* is the relative notch depth.

#### 2.5.2. Effective Fracture Toughness

Several adaptations of linear elastic fracture mechanics have been proposed to cover the nonlinear behavior of a material. One of them is the effective crack model (ECM) [4], which includes the effect of the pre-peak nonlinear behavior of a real concrete structure containing the initial notch through an equivalent elastic structure containing a notch of effective length *a*_e_ > *a*. The effective crack length *a*_e_ is calculated from the secant stiffness of the concrete specimen corresponding to the maximum load *F*_max_ and matching midspan deflection *d_F_*_max_. The value of *a*_e_ for the prismatic specimen with a central edge notch tested in the three-point bending configuration was determined according to [4] from the following relationship:(3)$$d_{F_{\max}}=\frac{F_{\max}}{4BE}{\left(\frac{S}{W}\right)}^{3}\left[1+\frac{5qS}{8F_{\max}}+{\left(\frac{W}{S}\right)}^{2}\left\{2.70+1.35\frac{qS}{F_{\max}}\right\}\;-\;0.84{\left(\frac{W}{S}\right)}^{3}\right]+\frac{9}{2}\frac{F_{\max}}{BE_{c}}\left(1+\frac{qS}{2F_{\max}}\right){\left(\frac{S}{W}\right)}^{2}F_{1}\left(\alpha_{e}\right),$$
where *E* is the static modulus of elasticity calculated from the initial part of the recorded *F−d* diagrams according to Stibor [20], *q* is the self-weight of the specimens per unit length, *S* is span length, and
(4)$$F_{1}\left(\alpha_{e}\right)\;=\overset{\alpha_{e}}{\underset{0}{\int }}xY^{2}\left(x\right)dx,$$
where *α*_e_ = *a*_e_/*W* is relative notch depth, and *Y*(*x*) is the function of geometry shown in Equation (2), where *α* is replaced by *α*_e_. Since the effective crack length *a*_e_ is expressed in Equation (4) as the argument of integral, the problem is solved using an iterative method.

Subsequently, the effective fracture toughness *K*_Ice_ value was calculated using a linear elastic fracture mechanics formula (Equation (1)), where *α* was replaced with *α_e_* in compliance with [4].

#### 2.5.3. Specific Fracture Energy

The complete *F−d* diagrams, including their post-peak parts, were employed to determine the work of fracture *W*_F_ value, which was given by the area under the diagram. In this case, *W*_F_ was calculated according to Stibor [20], where the area under the measured diagrams, the effect of the unmeasured part, and the self-weight of the specimen were considered. After that, the specific fracture energy *G*_F_ value was determined according to the RILEM method [21].
(5)$$G_{F}=\frac{W_{F}}{\left(W-a\right)B}.$$

The value of fracture energy was also investigated when the area under the curve was divided into two parts (see Figure 4). The first part *G*_F,1_ considers the area under the *F−d* diagram up to the maximum load *F*_max_, and the second part *G*_F,2_ considers the remaining area under the *F−d* diagram.

#### 2.5.4. The Double-K Fracture Model Parameters

The double-K fracture (2K) model [22] was used for the evaluation of the *F‒CMOD* diagrams to determine selected fracture parameters. This model allowed the calculation of the parameters describing different phases of the fracture process. The unstable fracture toughness *K*_Ic_^un^ is defined as the critical stress intensity factor corresponding to the maximum load *F*_max_, and it represents the phase of unstable crack propagation. This parameter is of similar meaning to the effective fracture toughness used in the ECM by Karihaloo [4]. The equivalent elastic crack length *a*_c_ was determined from the following equation [22]:(6)$$CMOD_{F_{\max}}=\frac{6F_{\max}Sa_{c}}{BW^{2}E}V\left(\alpha_{c}\right),$$
where *CMOD_F_*_max_ is the *CMOD* corresponding to maximum load *F*_max_, and
(7)$$V\left(\alpha_{c}\right)\;=0.76-2.28\alpha_{c}+3.87\alpha_{c}{}^{2}-2.04\alpha_{c}{}^{3}\frac{0.66}{{\left(1-\alpha_{c}\right)}^{2}},$$
where *α_c_* = (*a_c_* + *H*_0_)/(*W* + *H*_0_); *H*_0_ is the thickness of blades fixed on the bottom surface of the specimens between which the strain gauge was placed.

When the equivalent elastic crack length *a_c_* is known, *K*_Ic_^un^ was determined according to Equation (1), where *a_c_* was substituted by *a*, and the geometry function in this case was expressed as follows [4]:(8)$$Y\left(\frac{a_{c}}{W}\right)=\frac{1.99-\frac{a_{c}}{W}\left(1-\frac{a_{c}}{W}\right)\left(2.15-3.93\frac{a_{c}}{W}+2.70{\left(\frac{a_{c}}{W}\right)}^{2}\right)}{\left(1+2\frac{a_{c}}{W}\right){\left(1-\frac{a_{c}}{W}\right)}^{3/2}}.$$

The important parameter for nonlinear fracture mechanics calculation is the relationship between the stress and crack opening displacement (see Figure 5).

The fracture energy *G*_F_ is a derivative parameter of this relationship, which represents the area under this curve (softening function). There are two methods to obtain the parameters of the softening function. The first is based on the experimental determination of *G*_F_ from the uniaxial tensile strength test with deformation-controlled loading. The *G*_F_ is then calculated as the area under the *σ‒COD* diagram. However, it is quite hard to perform such a test in a stable way for concrete specimens, i.e., to also record the post-peak branch of the diagram. The other method consists of an indirect method of determination of *COD*_c_. In this case, *G*_F_ and *f*_t_, determined experimentally from the 3PBT and uniaxial tensile test, respectively, and a suitable shape of the softening function are the input parameters [23]. In the 2K model, the softening function has to be known to calculate the cohesive toughness at critical condition *K*_Ic_^c^, which can be interpreted as an increase in the resistance to crack propagation caused by the bridging of aggregate grains and other toughening mechanisms in the fracture process zone (FPZ) [22].

In this paper, the nonlinear softening function according to Hordijk [23], and *G*_F_ and *f*_t_ obtained by inverse analysis [24] were used for the calculation of related fracture parameters. The cohesive stress *σ*(*CTOD_c_*) at the tip of an initial notch at the critical state could be then obtained from this softening function.
(9)$$\sigma \left(CTOD_{c}\right)=f_{t}\left\{\left[1+{\left(c_{1}\frac{CTOD_{c}}{COD_{c}}\right)}^{3}\right]\exp\left(-c_{2}\frac{CTOD_{c}}{COD_{c}}\right)-\frac{CTOD_{c}}{COD_{c}}\left(1+c_{1}{}^{3}\right)\exp\left(-c_{2}\right)\right\},$$
where *f*_t_ is the tensile strength, *c*_1_ = 3 and *c*_2_ = 6.93 and are the material constants, which were taken from (Hordijk, 1991), and *CTOD_c_* is the critical crack-tip opening displacement according to Jenq and Shah [25] [Jeng 1985].
(10)$$CTOD_{c}=CMOD_{F_{\max}}{\left({\left(1-\frac{a}{a_{c}}\right)}^{2}+\;\left(1.081-1.149\frac{a}{W}\right)\left(\frac{a}{a_{c}}-{\left(\frac{a}{a_{c}}\right)}^{2}\right)\right)}^{\frac{1}{2}}.$$

*COD_c_* is the critical crack opening displacement calculated according to
(11)$$COD_{c}=\frac{5.136G_{F}}{f_{t}}.$$

The values of fracture energy *G*_F_ and tensile strength *f*_t_ were obtained by an inverse analysis based on an artificial neural network using the FraMePID-3PB Software [24]. The principle consists of the identification of the material parameters, which gives identical *F‒d* diagram responses to those obtained during real-time specimen loading. It is presumed that such strength is very close to the uniaxial tensile strength.

Subsequently, the linear function for the calculation of cohesive stress *σ*(*x*) along the length of the equivalent elastic crack can be formulated as follows:(12)$$\sigma \left(x\right)\;=\sigma \left(CTOD_{c}\right)+\frac{x-a}{a_{c}-a}\left(f_{t}-\sigma \left(CTOD_{c}\right)\right).$$

When this relation is known, the cohesive toughness *K*_Ic_^c^ is determined as follows:(13)$$K_{\mathrm{Ic}}^{c}=\overset{1}{\underset{a/a_{c}}{\int }}2\sqrt{\frac{a_{c}}{\pi }}\sigma \left(U\right)F\left(U,\frac{a_{c}}{W}\right)dU,$$
where the substitution *U* = *x*/*a_c_* is used, and *F*(*U, a_c_*/*W*) is determined according to [26] [Xu 1999].
(14)$$F\left(U,\frac{a_{c}}{W}\right)=\frac{3.52\left(1-U\right)}{{\left(1-\frac{a_{c}}{W}\right)}^{3/2}}-\frac{4.35-5,28U}{{\left(1-\frac{a_{c}}{W}\right)}^{\frac{1}{2}}}+\left(\frac{1.30-0,30U^{\frac{3}{2}}}{{\left(1-U^{2}\right)}^{\frac{1}{2}}}+0.83-1.76U\right)\left[1-\left(1-U\right)\frac{a_{c}}{W}\right].$$

The following formula based on the formerly obtained parameters was used to calculate the initial cracking toughness *K*_Ic_^ini^:(15)$$K_{\mathrm{Ic}}^{\mathrm{ini}}=K_{\mathrm{Ic}}^{\mathrm{un}}-K_{\mathrm{Ic}}^{c},$$
where *K*_Ic_^ini^ represents the phase of stable crack propagation.

Lastly, the load level *F*_ini_, which expresses the load at the outset of stable crack propagation from the initial notch, was determined according to
(16)$$F_{\mathrm{ini}}=\frac{4\cdot S_{M}\cdot K_{\mathrm{Ic}}^{\mathrm{ini}}}{S\cdot Y\left(\alpha \right)\cdot \sqrt{a}},$$
where *S_M_* is the section modulus (calculated as *S_M_* = 1/6∙*B*∙*W*^2^), *S* is the span length, and *Y*(*α*) is the geometry function (Equation (8)), where *α* = *a*/*W* is used instead of *a_c_*/*W*.

## 3. Results and Discussion

In this section, the results of the performed experiments are presented in tables and figures. Table 2 summarizes the characteristics of the air-void (A-V) system in the hardened non-frost-attacked concretes. These parameters merely serve as informative for this paper as the support for the interpretation of the related results. The results listed in Table 2 show basic and one of the most important differences between investigated concretes, namely, the total A-V content and the number of voids smaller than 300 microns were about 35% and 52% higher for concrete C1 compared to C2. This implies that concrete C1 should be more resistant to F–T than concrete C2. On the other hand, according to the paste–air ratio, the cement paste in concrete C2 should be denser and less permeable for the water medium.

Table 3 summarizes the mechanical and fracture characteristics of non-frost-attacked concretes. All characteristics were determined at the same time when the set exposed to 50 F–T cycles was tested. Comparing the results, including their variability, it can be stated that the basic mechanical characteristics such as dynamic modulus of elasticity and compressive strength were similar for both concretes. Moreover, the critical force for the start of unstable crack propagation (*F*_max_) was also very similar. Similar results could also be observed in the values of selected fracture characteristics, such as crack strength, fracture toughness, unstable fracture toughness, and effective crack extension. The difference between these parameters for C1 and C2 was up to 5%. The highest difference is recorded in the value of fracture energy (*G*_F_), which was about 14% higher for concrete C2. The values of *G*_F,1_ and *G*_F,2_ suggest that this difference was especially caused by the different post-peak behavior of investigated concretes; *G*_F,2_ was about 16% higher for C2, whereas *G*_F,1_ was almost the same for both concretes. A similar difference was recorded in the values of splitting tensile strength, which was about 15% lower for C2 compared to C1, but the variability for C2 was more than twofold higher. Similarly, the values of *F*_ini_ (critical force for the start of stable crack propagation), initial fracture toughness, and critical crack opening displacement (*COD_c_)* could not be simply compared because of the high differences in variability recorded for each concrete, which was about twofold higher (more than threefold for *COD_c_*) for C1 compared to C2.

The results of the F–T tests are presented in figures below. All parameters (except the changes in mass and dynamic Poisson’s ratio) are displayed as the relative values of the results obtained for frost and non-frost-attacked specimens of particular concrete as follows:(17)$$RV_{n}=\frac{P_{n}}{P_{0}},$$
where *RV_n_* is a relative value of a particular material characteristic determined for *n* F–T cycles (*n* = 0, 50, 100, and 200), *P_n_* is an average value of the set of specimens determined for a particular material characteristic after *n* F–T cycles, *P*_0_ is an average value of the set of non-frost-attacked specimens determined for the particular material characteristic (for *n* = 0; *RV_n_* = 1).

The error bars represent the relative standard deviation of the results for a particular set of specimens.
(18)$$RSD_{n}=RV_{n}\cdot CoV_{n},$$
where *RSD_n_* is a relative value of the standard deviation of a particular material characteristic determined for *n* F–T cycles (*n* = 0, 50, 100, and 200), *SSD_n_* is a sample standard deviation of the set of specimens determined for a particular material characteristic after *n* F–T cycles, and *CoV_n_* is a coefficient of variation of the set specimens determined after *n* F–T cycles.

The decrease in mass and almost constant value of dynamic Poisson’s ratio (see Figure 6) implies that the specimens are not significantly disturbed by macrocracks throughout the F–T test duration. As already indicated by other authors [27,28,29,30,31], the presence of significant cracks causes an increase in mass and changes in the values of Poisson’s ratio. It can be presumed that a small decrease in mass indicates desiccation of saturated test specimens during the freezing phase. A slow increase in mass recorded after 125 and 175 F–T cycles for C1 and C2, respectively, may indicate slow water uptake of the test specimens due to the existence of microcracks, which were also observed on the surface of the test specimens using a digital microscope (see Figure 7 and Figure 8). The water uptake led partially to the healing of existed or newly formed cracks. This behavior was observed for both concretes. In the case of C2, the width of observed surface cracks was higher than for C1, which implies that the existed cracks were not fully healed for C2 (see Figure 8), as observed for C1.

Although none of the concretes showed visible disruption, there were differences in the development of the dynamic modulus of elasticity and compressive strength (see Figure 9) for C1 and C2. It can be observed that concrete C1 exhibited better F–T resistance than concrete C2. The decrease in *E*_rL_ was about 5% and was quite stabilized after reaching 25 F–T cycles for C1. No decrease in the compressive strength was observed for C1. In both cases, the results exhibited low variability. The situation differed for concrete C2; a gradual decrease in *E*_rL_ and compressive strength up to about 20% was observed upon reaching 100 F–T cycles, after which the values of both parameters started to grow. The final decrease was about 15% and 4% for *E*_rL_ and compressive strength, respectively. The long-term experience of the authors with the utilization of the resonance method as a nondestructive technique for monitoring of the F–T damage in concrete suggests that a decrease in *E*_rL_ of about 15% indicates a decrease in the flexural or splitting tensile strength of at least about 25% [32,33]. This presumption is confirmed by the results presented in Figure 10a; the decrease in splitting tensile strength was about 40% for C2.

Figure 10 displays the changes in tensile characteristics of the investigated concretes during the F–T test. Two types of strength were determined: splitting tensile (in compliance with ČSN EN 12390-6 [34] on the fragments of specimens) and flexural strength determined on the notched specimens during fracture test (crack strength according to the terminology in the branch of fracture mechanics [35]). It can be stated that the results are rather contradictory. In the case of splitting tensile strength, a slight increase was observed for C1, while a gradual decrease of about 40% was recorded for C2 after 200 F–T cycles (see Figure 10a). This indicates that concrete C2 is not F–T-resistant according to the Czech standard [11]. On the other hand, a gradual decrease of about 16% after 100 F–T cycles for C1 and about 20% after 50 F–T cycles for C2 followed by an increase was observed for the crack strength. The crack strength was of the same value after 200 F–T cycles as before the start of freezing (see Figure 10b). It can be supposed that the differences in the development of tensile strengths originated in the loading regime, especially since the loading rate was extremely different. In the case of splitting tensile strength, the specimens are loaded by load increment at a rate of 0.05 MPa/s, whereas, during the fracture test, the specimens were loaded with a displacement increment at the rate of 0.02 mm/min.

The figures below display an advanced evaluation of the F–T damage based on the fracture mechanics of quasi-brittle materials. Note that the results of fracture tests are often accompanied by a high variability, which may in some cases reach more than 20%. The reason for this variability can be found in the test method itself. The method is based on the very slow loading of specimens; therefore, all microdefects are reflected in the resulting parameters. Another reason results from evaluation approaches, especially when the evaluation of the results is to a large extent based on the theoretical hypotheses, as is, e.g., the case in the double-*K* model. The assessment of fracture tests herein was based on an evaluation of *F−d* and also *F−CMOD* diagrams. The results based on the evaluation of *F−d* diagrams are presented first.

Specific fracture energy *G*_F_ (see Figure 11) is one of the most commonly used parameters for the assessment of the degree of F–T deterioration. The total fracture energy was calculated herein based on the *F−d* diagrams (see Section 2.5.3, Equation (5)). The results showed an increase in *G*_F_ of about 25% for C1 followed by a slight decrease after 50 F–T cycles. Nevertheless, the final value was about 12% higher than the value before the start of freezing. Similar findings were reported by Wardeh [36], who attributed this phenomenon to the presence of a microcrack network, which needs higher energy dissipation to complete fracture of the concrete. A slight increase of about 7% followed by a decrease with a final value of about 13% after 200 F–T cycles was observed for C2, which indicates an increase in brittleness with an increasing number of F–T cycles. Note that the value of *G*_F_ is strongly influenced by the area of the fracture surface. Commonly, a projection of the fractured ligament area is used for calculation, which can substantially influence the absolute value of *G*_F_. The actual fractured area can be more precisely specified by scanning the relief of the fracture surface using laser scanning techniques, which is labor- and time-consuming [37,38].

Figure 12 displays the total fracture energy divided into two parts (see Section 2.5.3, Figure 4); *G*_F,1_ represents the initial part of the energy consumed from the start of the fracture test up to the peak load, whereas *G*_F,2_ represents the post-peak part of the fracture energy. The results show a substantial increase (of about 60%) in the value of *G*_F,1_ for C1 after 50 F–T cycles, while almost the same value was recorded for C2 throughout the F–T test. According to the tensile behavior of concrete, as reported by Wardeh [36], the presence of a higher number of microcracks in C1 developed during the initial phase of loading could be confirmed, as also reflected in the post-peak behavior. However, the variability of the results was much higher for C1 than for C2 (see Figure 12a).

The energy *G*_F,2_ exhibited almost the same trend for both concretes with a different value of decrease at the end of the F–T test. The value of *G*_F,2_ increased by about 20% and 10% for C1 and C2, respectively, after 50 F–T cycles and was almost the same as before the start of freezing for C1, while a decrease of about 14% was observed for C2 after 200 F–T cycles.

Figure 13 displays the development of fracture toughness *K*_Ic_, determined according to the linear elastic fracture mechanics approach (Figure 13a) and effective fracture toughness *K*_Ice_, which includes the nonlinear behavior of concrete before reaching the peak load (Figure 13b). The trend of *K*_Ic_ development was the same as observed for the crack strength throughout the F–T test (see Figure 10b). This complies with the linear fracture mechanics approach [4]. A different trend was observed for *K*_Ice_ (see Figure 13b). The increase in this value was about 16% and even 25% after 50 and 200 F–T cycles, respectively, for concrete C1. On the other hand, a decrease of about 24% followed by a slow increase for concrete C2 was recorded after 50 F–T cycles. The final decrease in the value of *K*_Ice_ was about 8% for C2. Moreover, the effective crack extension increased for C1 (of about 40%) and decreased for C2 (of about 20%) throughout the F–T test (see Figure 14a). This indicates increasing nonlinearity caused by a higher number of microcracks along the FPZ before failure in C1 due to the F–T exposure. According to the results, it can be stated that concrete C2 became more brittle due to exposure to F–T cycles compared to concrete C1.

Other fracture parameters were obtained from the *F−CMOD* diagrams evaluated using the double-*K* fracture model, which allowed an analysis of different phases of the fracture process. Note that the tensile strength needed for the estimation of the softening function was in this case obtained via an indirect method (see Section 2.5.4). The development of the tensile strength identified according to [24] for a particular test set and concrete is displayed in Figure 14b. The results obtained based on the identification presumed a decrease in the value of tensile strength by about 20% for the specimens subjected to 50 F–T cycles for both concretes. A re-increase in tensile strength was expected after 100 and after 50 F–T cycles for C1 and C2, respectively. In the case of C2, it was expected that the tensile strength would be about 20% higher after 200 F–T cycles than that estimated for the specimens before the start of freezing. The increasing trend of development identified for specimens subjected to 100 and 200 F–T cycles did not correspond to the trend of crack strength development obtained from fracture tests for C2 concrete (see Figure 10b). The trend of tensile strength development identified for C1 was in good agreement with the trend of crack strength development (see Figure 10b).

Unstable fracture toughness *K*_Ic_^un^ and cohesive fracture toughness *K*_Ic_^c^ are two basic fracture parameters determined using the *F−CMOD* diagrams. The cohesive fracture toughness *K*_Ic_^c^, as a component of unstable fracture toughness *K*_Ic_^un^, reflects the cohesive mechanisms in the FPZ. Many micro-failure mechanisms such as matrix microcracking, debonding of the cement–matrix interface, crack deflection, grain bridging, and crack branching, which consume energy during the crack propagation, are responsible for the stress transfer [22]. If the component of cohesive fracture toughness *K*_Ic_^c^ is subtracted from the unstable fracture toughness, the value of initiation fracture toughness *K*_Ic_^ini^ is obtained. The critical values of the fracture toughness are obtained at the load level *F*_max_, upon reaching the equivalent elastic crack extension and critical crack-tip opening displacement (*CTOD*_c_).

Figure 15a shows an increase in value of equivalent elastic crack extension (calculated using Equation (6)) of about 50% followed by a drop after 100 F–T cycles for C1. Concerning the C2 concrete, an increase of about 24% was followed by a steep drop observed after 50 F–T cycles. The final values recorded after 200 F–T cycles were about 30% higher and 12% lower for C1 and C2, respectively, when compared to the values before the start of freezing. The increase in equivalent elastic crack extension indicates an increasing nonlinear behavior before the failure caused by an increasing number of microcracks in the material due to the F–T action [34]. Simultaneously, more energy was consumed to completely break the material (see Figure 11 or Figure 12); thus, it behaved more ductile.

Figure 15b shows an increase in the value of *CTOD*_c_ of more than 60% for C1 after 200 F–T cycles. This indicates an increase in the fictitious crack width before the failure due to F–T exposure. An increase of about 34% followed by a steep decrease of about 50% was recorded for C2 after 50 F–T cycles. This indicates a gradual increase in brittleness of C2 with an increasing number of F–T cycles.

The above results correspond to the results of fracture toughness obtained using the 2K model. As shown in Figure 16 or Figure 17a, an increase of about 20% was recorded for C1 concrete for cohesive (*K*_Ic_^c^), unstable (*K*_Ic_^un^), and initial fracture (*K*_Ic_^ini^) toughness after 50 F–T cycles, after which all toughness components were already stabilized during the remainder of the F–T test. In the case of concrete C2, the trend of development slightly differed for the three toughness components. No decrease in *K*_Ic_^c^ was recorded after 50 F–T cycles, while a gradual and steep decrease was recorded for *K*_Ic_^un^ and *K*_Ic_^ini^, respectively. A maximum decrease of about 11% and 17% followed by an increase in the value of cohesive and unstable fracture toughness, respectively, was observed after 100 F–T cycles. The final value of cohesive fracture toughness was about 8% higher compared to the value before the start of freezing.

The increase in cohesive fracture toughness recorded for C1 reflects the action of the cohesive forces in the FPZ, which led to softening of the material [36], and it indicates the energy absorbed by the cohesive stresses acting on the fictitious crack during the stable crack propagation [39].

The descending trend of *K*_Ic_^un^, *K*_Ic_^ini^, and *K*_Ic_^c^ indicates higher deterioration of C2 accompanied by more brittle failure due to the F–T action compared to C1 concrete.

Figure 17b shows the development of the *F*_ini_/*F*_max_ ratio for both concretes during the F–T test. The results show a gradual increase of up to 27% for C1 after 100 F–T cycles. The final value after 200 F–T cycles was about 16% higher than before the start of freezing. A gradual decrease of up to 15% was recorded for C2 after 200 F–T cycles. The increasing load ratio indicates an extension of the linear part of the diagram, which expresses the later onset of the stable crack propagation, i.e., the resistance to the crack onset increased during the F–T exposure for C1 concrete until reaching 100 F–T cycles.

All these parameters together indicate an enhanced resistance of C1 concrete to brittle fracture.

AE signals were continuously recorded throughout the fracture tests to evaluate the extent of the deterioration of the concrete specimens due to exposure to F–T cycles. The changes in the RMS value were evaluated for selected load levels *F*_ini_ and *F*_max_. The changes in loading force at the selected load levels are displayed separately in Figure 18. The main changes in the values of *F*_ini_ were recorded for both concretes after 50 F–T cycles. After that, the value of *F*_ini_ was almost stabilized for both concretes. Concerning *F*_max_, a gradual decrease was observed for both concretes up to 100 F–T cycles, after which the value of *F*_max_ increased for C1 and C2. The variability in the loading forces was reflected in the evaluation of the RMS values, which was determined for the region of the average value of *F*_ini_ ± standard deviation. The same procedure was applied for the load level *F*_max_ and region 0–*F*_max_.

It is important to emphasize that AE results are influenced by two different loading processes which are concurrently in progress. The first is the F–T action, which proceeds continuously, and the second one is the fracture test performed after a selected number of F–T cycles. Considering that the F–T cycles act as a continuous loading and unloading process, failure occurs due to the progressive damage of the internal structure of concrete, which is reflected in a decrease in the acoustic signal amplitude or a shifting of the dominant frequency, as reported in [40]. This irreversible damage is then reflected in the AE parameters recorded during the fracture tests. It is supposed that a decrease in the strength of the AE signal is lower for specimens with a higher degree of internal disruptions.

The results of AE show a gradual decrease in RMS values for both concretes at all investigated load levels (see Figure 19). However, a higher decrease was in all cases recorded for C2 concrete. This indicates that the internal structure of C2 concrete was strongly disturbed with an increasing number of F–T cycles, during which the growth and coalescence of existing microcracks and the formation of new microcracks occurred. A lower acoustic response is recorded for disturbed structural bonds compared to those that are non-disturbed. A gradual decrease in RMS of about 40% and 70% for C1 and C2, respectively, was recorded at the load level *F*_ini_ (expected crack initiation) after 200 F–T cycles. Almost the same descending trend of RMS for C1 was recorded when evaluated for the load range of 0 up to the *F*_max_. The increasing deterioration of C2 was reflected in a steep decrease in RMS, the value of which was negligible compared to the non-frost-attacked specimens after 200 F–T cycles. Although the resonance test showed a resistance of C1 concrete to the F–T action, the AE measurement revealed a gradual degradation of internal structure, as reflected in a decrease in RMS value of about 40% after 200 F–T cycles. This proved the evolution of the microcrack network during the pre-peak loading phase, which was reflected in the increase of the fracture toughness, crack extension, and fracture energy.

The presented results are only a small part of an extensive experimental investigation focused on the F–T damage of concrete tested in the laboratory and on-site. The results of the performed experimental–numerical analysis indicate the high potential of employing fracture mechanics as a tool for the assessment of F–T damage, especially if the double-K model is also employed. The performed investigation also investigates the issue of methods for inverse analysis which might be adjusted to the pre-cracked material (due to the F–T action) submitted to the fracture tests. This novel approach provides a wider range of evaluated fracture parameters than usually presented in this context; thus, it is not possible to fully support the discussion with the literature. However, we believe that it could build a base for further investigation and a comparison of results with other research groups.

## 4. Conclusions

This paper presented the results of an experimental program aimed at the assessment of F–T resistance of concrete based on fracture test evaluation, accompanied by acoustic emission measurement. Two concretes of similar mechanical characteristics were manufactured for the experiment. The main differences between the C1 and C2 concrete were in the total number of air voids and in the parameter A_300_, which were both higher for C1. It is important to emphasize that both concretes did not exhibit macro-defects throughout the F–T test duration, i.e., no surface scaling or macrocracks were observed.

The evaluation of the fracture characteristics was performed on the basis of experimentally recorded *F*–*d* and *F*–*CMOD* diagrams using two different approaches: linear fracture mechanics completed with the effective crack model and double-K model. It was observed that both approaches gave similar results, especially if the nonlinear behavior before the peak load was considered.

The changes in the root-mean-square (RMS) of the acoustic emission signal were evaluated from the continuous AE measurement.

According to the results, the following conclusions can be drawn:It can be supposed that C1 concrete exhibited better resistance to the F–T action compared to C2. All fracture parameters together indicated an enhancing resistance of C1 concrete to brittle fracture during the F–T test.It can be stated that the continuous AE measurement is beneficial for the assessment of the extent of concrete deterioration and suitably supplements the fracture test evaluation.The results showed that the F–T damage was more reflected in the fracture toughness parameters than in the fracture energy.The F–T damage of the investigated concretes was reflected in the value of fracture energy, which increased with an increase in the microcrack network and decreased for concrete with a more seriously damaged structure. To confirm the presence of microcracks, it seems to be beneficial to calculate the fracture energy G_F,1,_ and G_F,2_ separately for pre- and post-peak load phases. The presence of microcracks led to an increase in the pre-peak fracture energy G_F,1_ (see Figure 12a). It can be stated that an increase in *G*_F,1_ for concrete C1 was caused especially by softening in the FPZ, as reflected by the increase in the value of effective and unstable fracture toughness (see Figure 13b or Figure 16b) and in the post-peak behavior.It can be stated that the F–T damage was notably reflected in the characteristics of the fictitious crack represented herein by the effective crack extension and critical crack-tip opening displacement. Both parameters indicate the ductility/brittleness of the material. According to the results, it can be supposed that an increase in crack extension and opening indicates increasing nonlinear behavior before failure, implying an increase in ductility of C1 during F–T exposure. On the other hand, the C2 became more brittle with an increasing number of F–T cycles (see Figure 14a or Figure 15).The double-*K* model seems to be beneficial for the evaluation of F–T damage because it enables distinguishing the different phases of crack propagation. Additionally, it provides the possibility to calculate the cohesive component of the fracture toughness, which represents the action of cohesive forces along the fictitious crack and indicates the risk of brittle fracture.Comparing the results of fracture tests with the resonance method and splitting tensile strength test, it can be stated that all testing methods gave the same conclusion, i.e., C1 concrete is more F–T-resistant than C2. However, the fracture test evaluation provided more detailed information about the internal structure deterioration due to F–T exposure.The decrease in fracture parameters of C2 concrete corresponded well to the decrease in dynamic modulus of elasticity (see Figure 9a) recorded during the F–T test. Unfortunately, there are no criteria for related damage factors determined by the Czech standard. It can be supposed that the microcracks indicated by the fracture parameters for C1 were reflected by a slight decrease in its dynamic modulus (about 5%) determined by the resonance method. However, it is not possible to assess the ductility or brittleness using the resonance method.The main disadvantages of the fracture test performed in the context of F–T resistance are the time consumption (one test lasts at least 40 min), labor intensiveness, and the process of evaluation, which limit its wider utilization in standard practice.

## Figures and Tables

**Figure 1 materials-14-06378-f001:**
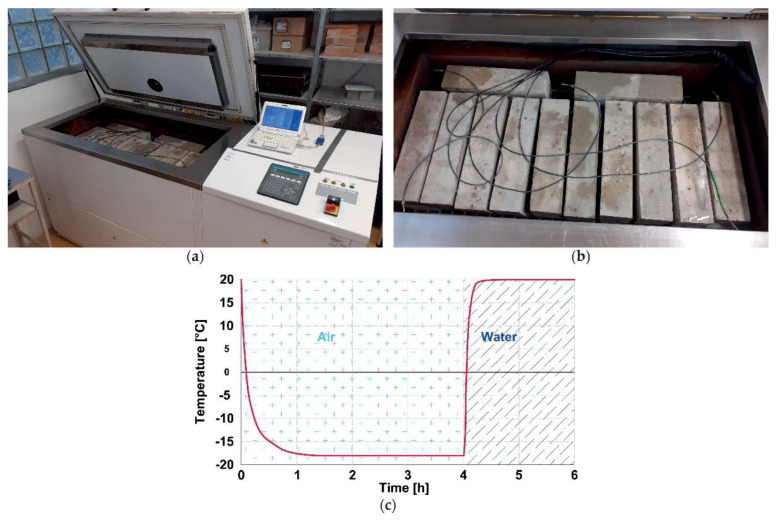
Arrangement of freeze–thaw test: (**a**) automatic freeze–thaw cabinet KD 20; (**b**) detail of specimens arrangement during F–T test; (**c**) one F–T cycle.

**Figure 2 materials-14-06378-f002:**
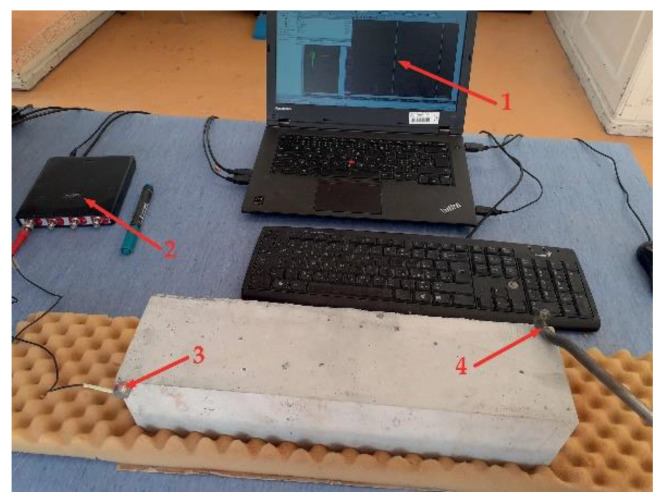
Measurement of resonant frequencies (1—computer equipped with software for determination of resonant frequencies; 2—Handyscope HS4 oscilloscope; 3—acoustic sensor; 4—impact hammer).

**Figure 3 materials-14-06378-f003:**
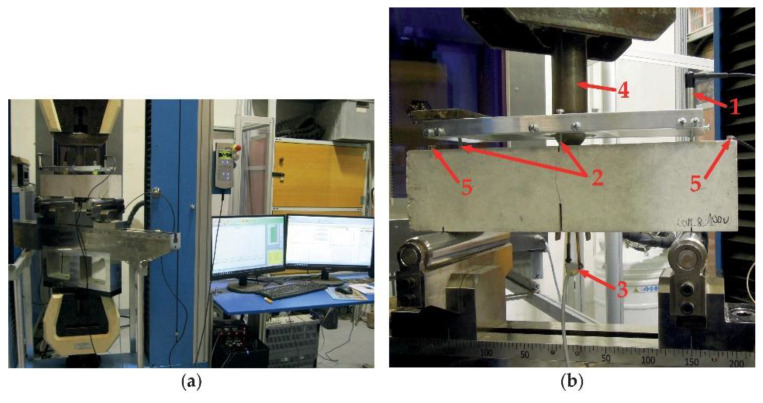
(**a**) Arrangement of fracture tests; (**b**) details of the test specimens and sensors (1—inductive sensor in a frame; 2—rectifying screws in the middle of the span and above the support; 3—strain gauge between the blades; 4—applied load; 5—acoustic emission sensors).

**Figure 4 materials-14-06378-f004:**
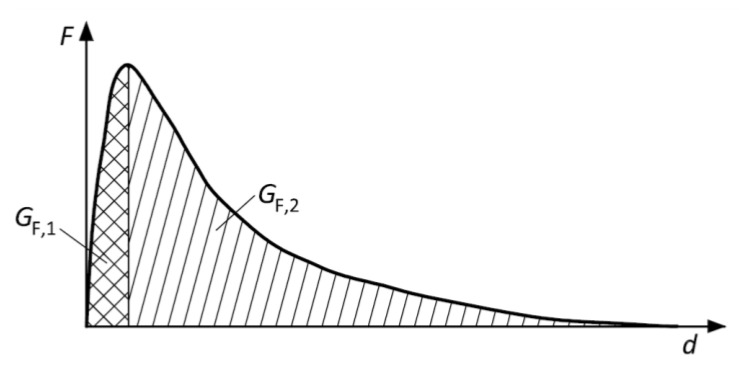
The considered parts of fracture energy.

**Figure 5 materials-14-06378-f005:**
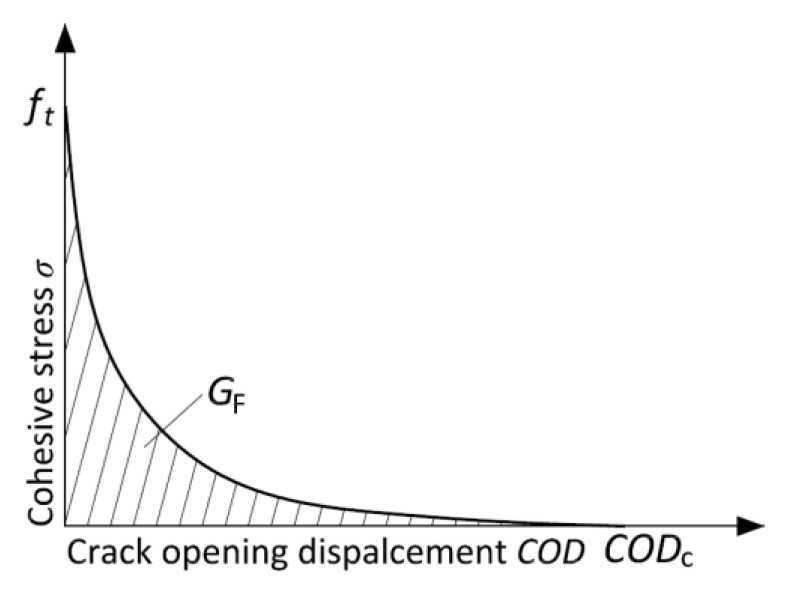
The parameters of softening function.

**Figure 6 materials-14-06378-f006:**
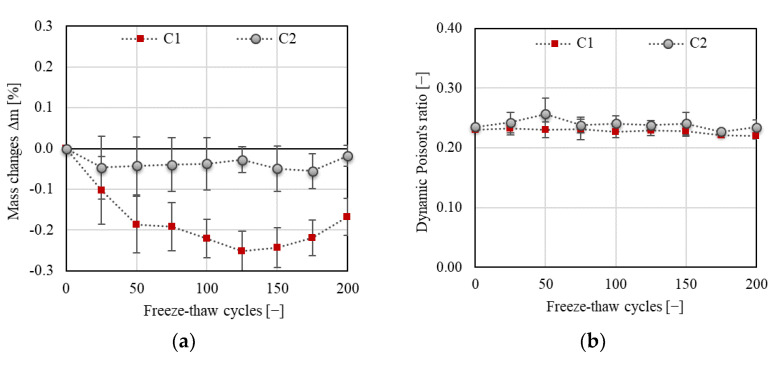
(**a**) Changes in mass; (**b**) Poisson’s ratio.

**Figure 7 materials-14-06378-f007:**
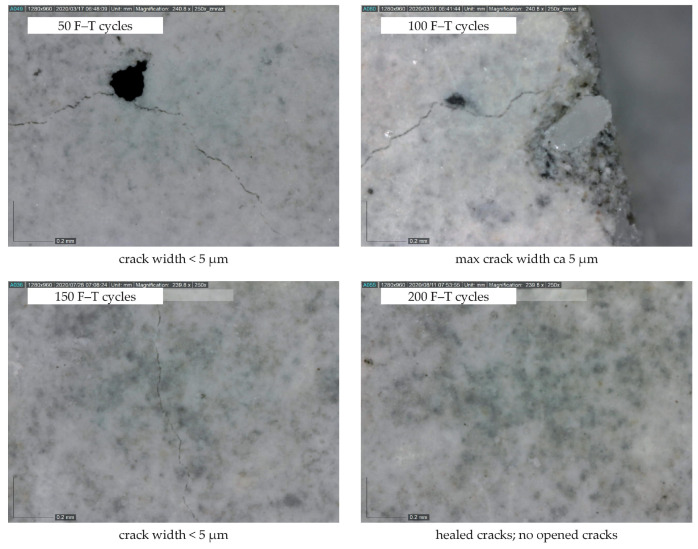
Microcracks on the specimen surface (C1; magnification 250×).

**Figure 8 materials-14-06378-f008:**
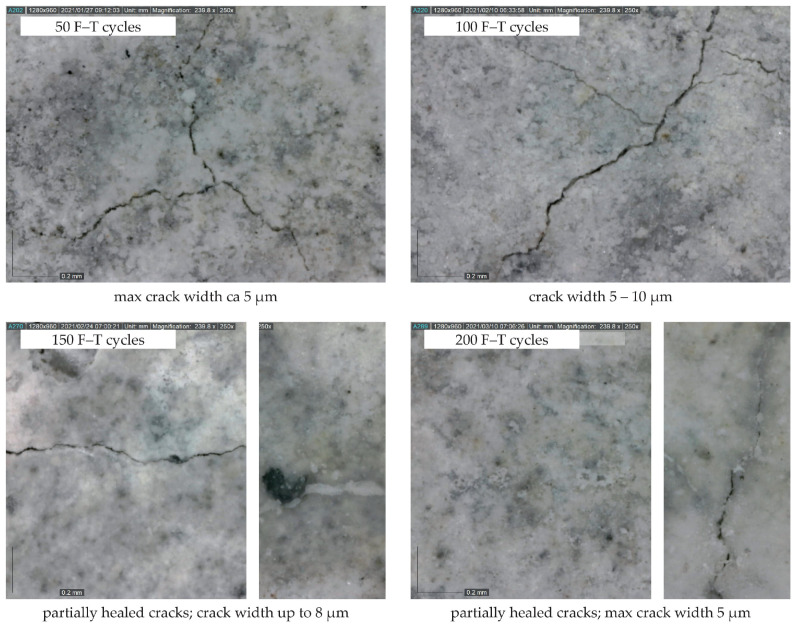
Microcracks on the specimen surface (C2; magnification 250×).

**Figure 9 materials-14-06378-f009:**
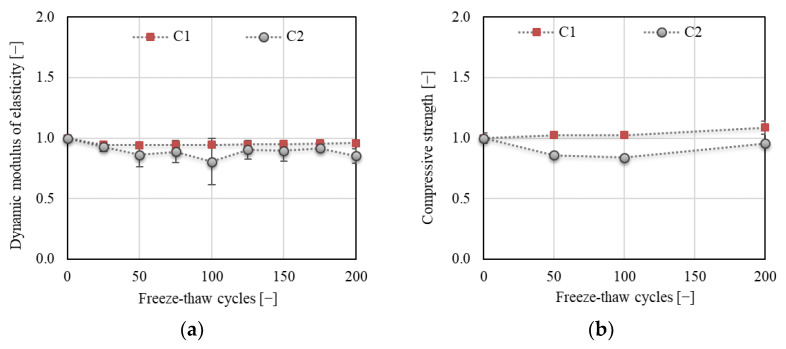
(**a**) Dynamic modulus of elasticity (*E*_rL_); (**b**) compressive strength.

**Figure 10 materials-14-06378-f010:**
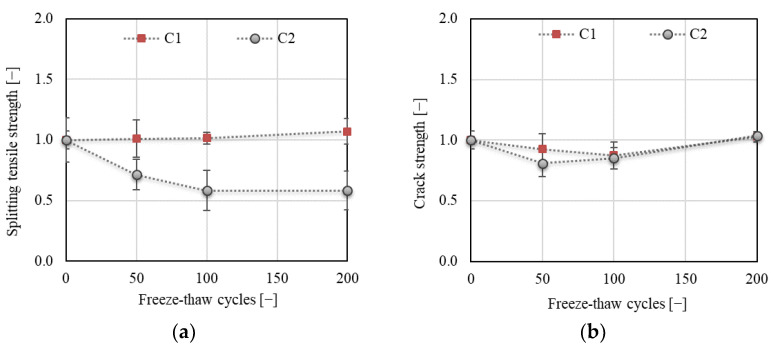
(**a**) Splitting tensile strength; (**b**) crack strength.

**Figure 11 materials-14-06378-f011:**
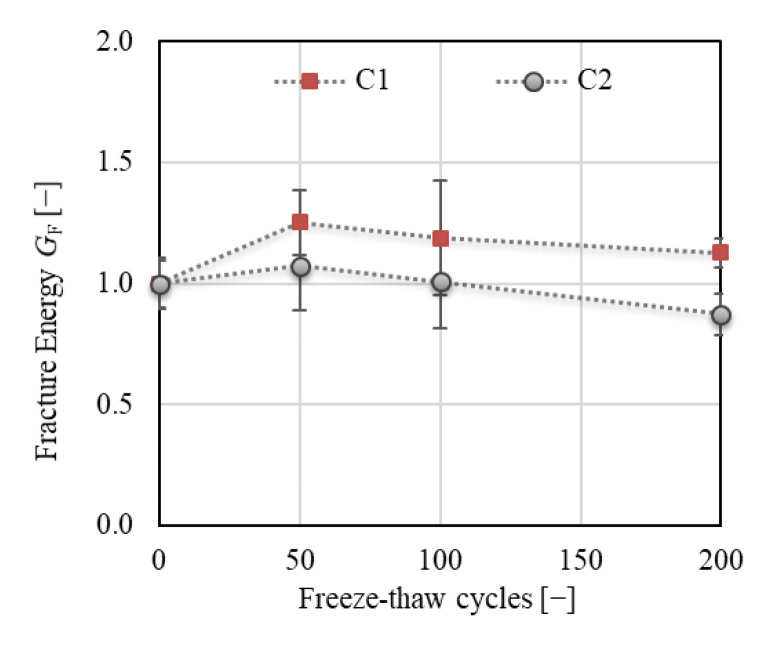
Specific fracture energy *G*_F_.

**Figure 12 materials-14-06378-f012:**
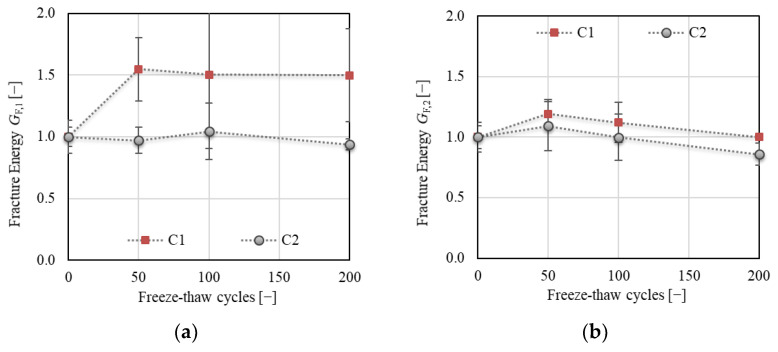
(**a**) Fracture energy *G*_F,1_ (up to the peak load); (**b**) fracture energy *G*_F,2_ (post-peak part).

**Figure 13 materials-14-06378-f013:**
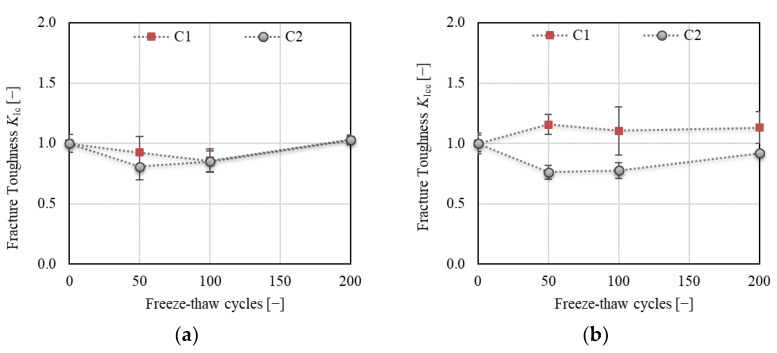
(**a**) Fracture toughness *K*_Ic_; (**b**) effective fracture toughness *K*_Ice_.

**Figure 14 materials-14-06378-f014:**
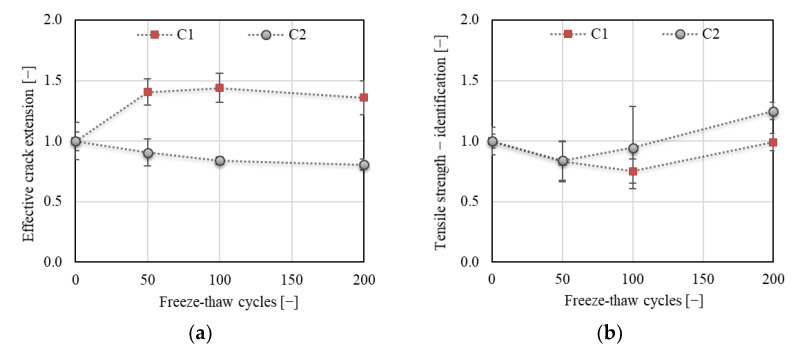
(**a**) Effective crack extension; (**b**) tensile strength from identification.

**Figure 15 materials-14-06378-f015:**
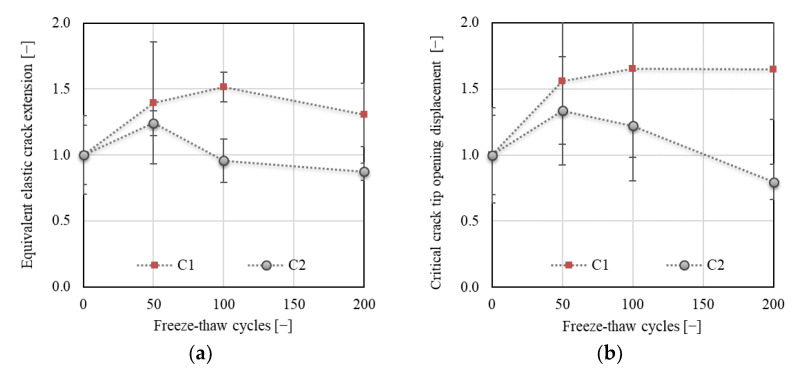
(**a**) Equivalent elastic crack extension; (**b**) critical crack-tip opening displacement (CTOD).

**Figure 16 materials-14-06378-f016:**
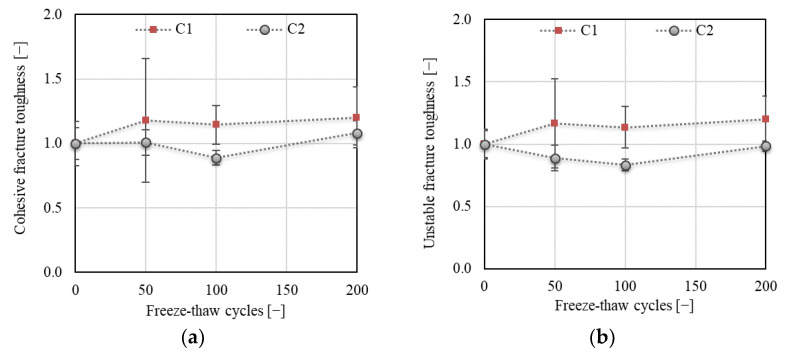
(**a**) Cohesive fracture toughness *K*_Ic_^c^; (**b**) unstable fracture toughness *K*_Ic_^un^.

**Figure 17 materials-14-06378-f017:**
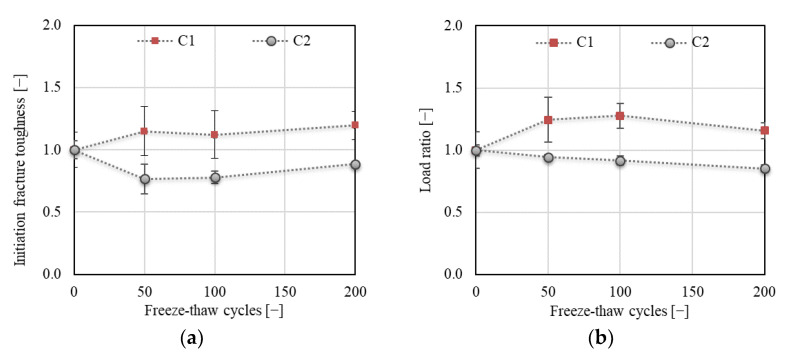
(**a**) Initiation fracture toughness *K*_Ic*ini*_; (**b**) load *F*_ini_/F_max_ ratio.

**Figure 18 materials-14-06378-f018:**
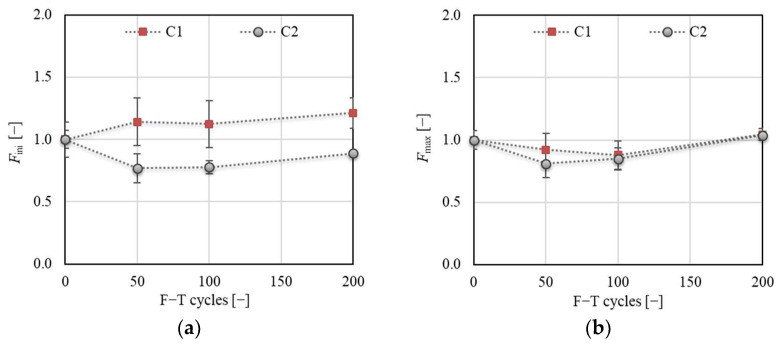
(**a**) Load level *F*_ini_; (**b**) load level *F*_max_.

**Figure 19 materials-14-06378-f019:**
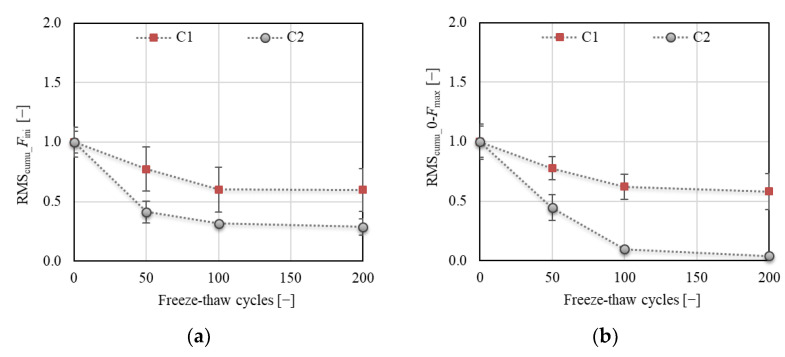
Acoustic emission: (**a**) RMS_cumu_ for *F*_ini_; (**b**) RMS_cumu_ for interval 0–*F*_max_.

**Table 1 materials-14-06378-t001:** Composition of C1 and C2 concrete in kg per 1 m^3^ of fresh concrete and basic characteristics of fresh concretes.

Components/Characteristics	C1	C2
Cement CEM I 42.5 R	390	390
Sand 0–4 mm (Tovačov, CZ)	810	810
Gravel 4–8 mm (Luleč, CZ)	160	160
Gravel 8–16 mm (Olbramovice, CZ)	760	760
Admixture Sika ViscoCrete-4035	1.00	0.40
Air-entraining admixture LPS A 94	0.55	0.20
Admixture Sika ViscoFlow-25	1.60	0.64
Water	178	198
w/c	0.46 (0.43 *)	0.51 (0.47 *)
Density of fresh concrete (kg/m^3^)	2290	2340
Air content (%)	4.3–5.0	2.1–2.5
Workability (flow-table test) (mm)	420/430	410/420

* Value after subtraction of the admixtures and aggregate absorption.

**Table 2 materials-14-06378-t002:** Air-void system of hardened concretes: average value (standard deviation).

Parameter	C1	C2
Total air-void content (%)	4.26 (0.372)	2.77 (0.127)
Specific surface (mm^−1^)	24.4 (2.74)	23.0 (1.56)
Paste–air ratio	7.22 (0.64)	11.75 (0.54)
Spacing factor (mm)	0.23 (0.019)	0.30 (0.026)
A_300_ (%)	1.31 (0.048)	0.63 (0.014)

**Table 3 materials-14-06378-t003:** Mechanical, fracture, and AE characteristics of non-frost attacked concretes C1 and C2: average value (standard deviation).

Parameter	C1	C2
Dynamic modulus of elasticity (GPa)	43.330 (0.976)	42.980 (0.727)
Compressive strength * (MPa)	60.0 (0.1)	57.0 (2.6)
Splitting tensile strength * (MPa)	5.41 (0.4)	4.61 (0.85)
Crack strength (MPa)	5.02 (0.16)	5.35 (0.40)
Tensile strength, identification (MPa)	3.20 (0.37)	2.99 (0.21)
Load level *F*_ini_ (kN)	3.41 (0.59)	3.95 (0.36)
Maximum load *F*_max_ (kN)	5.13 (0.13)	5.31 (0.40)
Effective fracture toughness (MPa.m^1/2^)	1.249 (0.105)	1.371 (0.093)
Fracture toughness (MPa.m^1/2^)	0.773 (0.022)	0.823 (0.062)
Fracture energy G_F_ (J/m^2^)	127.7 (12.33)	146.0 (15.5)
Fracture energy G_F,1_ (J/m^2^)	22.2 (3.00)	23.4 (1.88)
Fracture energy G_F,2_ (J/m^2^)	105.5 (9.71)	122.7 (14.89)
Initial fracture toughness (MPa.m^1/2^)	0.520 (0.09)	0.619 (0.056)
Unstable fracture toughness (MPa.m^1/2^)	1.225 (0.145)	1.205 (0.128)
Effective crack extension (mm)	16.91 (2.6)	17.83 (1.4)
Equivalent crack extension (mm)	15.56 (3.5)	13.00 (3.9)
Critical crack tip opening displacement (mm)	0.0244 (0.003)	0.0243 (0.007)
RMS_cumu__F_max_ (mV)	0.1663 (0.0405)	0.124 (0.0182)

* Determined on the fragments of fractured specimens.

## Data Availability

The data presented in this study are available on request from the corresponding author.

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
