# Peer review of "Advanced Evaluation of the Freeze–Thaw Damage of Concrete Based on the Fracture Tests"

_materials, 2021, doi:10.3390/ma14216378_

Round 1

Reviewer 1 Report

The article “Advanced evaluation of the freeze–thaw damage of concrete based on the fracture tests and acoustic emission measurement” can be a good addition to body of knowledge available on the subject if minor corrections are made as suggested below:

  • The authors need to define acronyms in abstract.
  • Please provide a list of nomenclature at the start of paper.
  • Add some percentage differences in abstract for C1 and C2.
  • Justify the use of Super-plasticizer Sika ViscoCrete-4035 and Viscoflow 25 CZ with reasons and references.
  • Authors need to put some recommendations and future research directions based on their findings.
  • The conclusion section seems quite long, if possible could you please shorten it with the only required information.
  • The results should be supported with literature or the statement of novelty must be provided.
  • is ČSN 73 1322 standard a new standard? as for CEN/TS 12390-9:2016, the duration of 1 FT cycle is 24 hours and it is an European standard. 
  • The number of references should be enhanced up to fifty. Authors can cite following articles:
  1. Jhatial, A.A., Kancir, I.V. & Serdar, M. (2021). “Comparative Study of Selected Properties of Three Binders: Blended Portland Cement, Calcium Sulfoaluminate Cement And Alkali Activated Material Based Concrete”, 2nd International Conference on Construction Materials for Sustainable Future, Bled, Slovenia, 27-28 April 2021.
  2. Jhatial, A.A., Serdar, M. & Ye, G. (2020). “Review on concrete under combined environmental actions and possibilities for application to alkali activated materials”, 6th Symposium on Doctoral Studies in Civil Engineering, Faculty of Civil Engineering, University of Zagreb, Croatia, 7-8 September 2020. DOI: 10.5592/CO/PhDSym.2020.09.
  3. Wang. D. et al. (2017) Durability of concrete containing fly ash and silica fume against combined freezing-thawing and sulfate attack. https://doi.org/10.1016/j.conbuildmat.2017.04.172
  4. Zhang W. et al. (2020). Influence of damage degree on the degradation of concrete under freezing-thawing cycles. https://doi.org/10.1016/j.conbuildmat.2020.119903

All in all, the article has potential and can contribute significantly to the scientific pool of freezing thawing of concrete.

Author Response

First of all, we would like to thank you for the time you spent on review of our manuscript. The responses to your comments are implemented in the revised manuscript and also listed separately in the file attached.

Reviewer 2 Report

The article is devoted to evaluation of the freeze-thaw damage of concrete based on the fracture test and acoustic emission measurement. After a rather short introduction, a fairly large part of the article is taken up by a description of how the tests were carried out and the numerous parameters calculated on the basis of the obtained results and assumptions. At this stage the article is still interesting for the reader who expects interesting comparisons of the above mentioned parameters in the results section and their discussion.

However, the third chapter is boring. Successive calculated parameters, with few exceptions, lead to very similar conclusions, although formulated differently and on different basis. Repeated, almost indistinguishable from each other, successive diagrams of calculated parameters of two (sic!) series of concrete of slightly different composition get boring about halfway through the chapter. Almost half of the calculated and analysed parameters could be thrown out of the article without any significant loss. Instead, the authors could be tempted to make cross comparisons of various combinations of calculated parameters in order to show their correlation or, on the contrary, lack thereof. I also miss more content on acoustic emission in the article. Mentioned in the title, it would seem to be one of the two "legs" on which the article is founded. In the meantime, it is at best a "pseudopodium", to which about one page is devoted at the end of the article, and only two out of a multitude of drawings. It would be good, for illustrative purposes only, to include at least one graph showing the raw results obtained during the acoustic emission test.

Apart from these general remarks, I am unclear as to whether all samples were subjected to tests of fundamental longitudinal frequency before the freeze-thaw cycles began, or only selected reference samples? It would be worthwhile to clarify this in the article. Furthermore, in my opinion the graphs in the article have the disadvantage that the place where each graph starts, i.e. the point on the vertical axis corresponding to value 1, is not labelled due to the adopted scale of values on this axis. I find this striking, but perhaps I am alone in this opinion.

To sum up, I do not have any comments on the way the tests were performed and the interpretation of their results, although I believe that it would have been possible to perform these tests on a larger number of more diverse series. I consider the article worthy of publication after making changes, the extent of which I leave to the authors' discretion. Only a small part of the weaknesses I have described in the article I consider necessary to eliminate. I think the authors will be able to identify them. The remaining changes would certainly make the article more interesting, but they are not an obstacle to its publication.

Author Response

(The authors gave the same response as above.)

Reviewer 3 Report

The paper is interesting, topic is presented very well. Whole research is presented very widely. Authors used and described two methods for evaluation of concrete damage which is good base for further research of that topic which is very important for serviceability of concrete structures in aggressive environment.

Main question addressed by the research is evaluation of the freeze-thaw damage of concrete. The evaluation is made by two advanced methods, fracture test, based on fracture mechanics and acoustic emission measurement. The test consists of two types of concrete specimens with different amount of air voids and A300 parameter. This topic is relevant which can be seen from literature review, and it is original because of used test methods. First method is destructive and cannot be used on real structures, but second method is non-destructive, and it can be used for onsite testing. The topic of the paper is continuation the research in this field as it can be seen from introduction of the paper. The paper is written well, it is very clear and easy to read. The results are presented in detail and comment and conclusions are clear and they are consistent with the presented results.

Author Response

We would like to thank you for the time you spent on review of our manuscript.